# Unraveling the *Tropaeolum majus* L. (Nasturtium) Root-Associated Bacterial Community in Search of Potential Biofertilizers

**DOI:** 10.3390/microorganisms10030638

**Published:** 2022-03-17

**Authors:** Isabella Dal’Rio, Jackeline Rossetti Mateus, Lucy Seldin

**Affiliations:** Laboratório de Genética Microbiana, Departamento de Microbiologia Geral, Instituto de Microbiologia Paulo de Góes, Centro de Ciências da Saúde, Universidade Federal do Rio de Janeiro, Bloco I, Ilha do Fundão, Rio de Janeiro CEP 21941-902, Brazil; dalrio@micro.ufrj.br (I.D.); jacky.rossetti@micro.ufrj.br (J.R.M.)

**Keywords:** *Tropaeolum majus*, nasturtium, bacterial community, root, plant growth-promoting bacteria, biofertilizer

## Abstract

Although *Tropaeolum majus* (nasturtium) is an agriculturally and economically important plant, especially due to the presence of edible flowers and its medicinal properties, its microbiome is quite unexplored. Here, the structure of the total bacterial community associated with the rhizosphere, endosphere and bulk soil of *T. majus* was determined by 16S rRNA amplicon metagenomic sequencing. A decrease in diversity and richness from bulk soil to the rhizosphere and from the rhizosphere to the endosphere was observed in the alpha diversity analyses. The phylum Proteobacteria was the most dominant in the bacteriome of the three sites evaluated, whereas the genera *Pseudomonas* and *Ralstonia* showed a significantly higher relative abundance in the rhizosphere and endosphere communities, respectively. Plant growth-promoting bacteria (236 PGPB) were also isolated from the *T. majus* endosphere, and 76 strains belonging to 11 different genera, mostly *Serratia*, *Raoultella* and *Klebsiella*, showed positive results for at least four out of six plant growth-promoting tests performed. The selection of PGPB associated with *T. majus* can result in the development of a biofertilizer with activity against phytopathogens and capable of favoring the development of this important plant.

## 1. Introduction

*Tropaeolum majus*, popularly known as garden nasturtium or nasturtium, is an edible plant native to South America that belongs to the Tropaeolaceae family. Recently, an increase in agricultural and economic interest in this plant has been observed because of its several medicinal characteristics and the presence of colorful edible flowers that can compose landscaping projects and/or be used for cooking. As another beneficial feature, *T. majus* is easily adaptable to abiotic stresses, such as temperature fluctuations and low soil fertility [1,2].

The flowers, leaves and stems of *T. majus* can be used in fresh salads, the seeds can be pickled, and even the roots can be used in tea consumption [2,3]. The flowers of *T. majus* are a good source of macroelements and microelements, such as potassium, phosphorus, calcium and magnesium, zinc, copper and iron [2]. As a low-maintenance and highly nutritious plant, *T. majus* can be an alternative to socioeconomically deprived populations [4]. However, their consumption and usage are not widespread in many countries.

The presence of a spicy flavor in *T. majus* plants is due to the presence of glucosinolates, especially isothiocyanates, which are bioactive compounds that exhibit a fast and long-lasting antitumoral effect in vitro [5,6]. In addition, plant extracts also show anti-inflammatory, antibiotic, anthelmintic and antioxidant properties [5,7,8]. They are mostly composed of fatty acids, such as oleic and linoleic acids, and phenolic compounds, such as flavonoids [8]. The flavonoid isoquercitrin is one of the most explored components in *T. majus* extracts, as this compound induces diuresis and is a possible treatment for hypertension reduction and cardiovascular diseases [9].

Alternative agricultural methods, such as organic and agroforestry systems, may also benefit from using *T. majus*. Intercropping using corn (*Zea mays*) and *T. majus* is able to maintain corn yield and biomass quality, while *T. majus* flowers work as pollinators and may confer profit due to the commercialization of its derivatives [10]. In addition, *T. majus* can help adjacent plants, acting as a plague repellent and enhancing soil fertility [11]. Thus, *T. majus* usage is especially important to organic agriculture since it reduces the need for pesticides while favoring the growth of adjacent plants.

However, like other plants, *T. majus* may be susceptible to some phytopathogens, such as mosaic viruses [12,13] and fungal species associated with anthracnose [14]. Natural infection of *T. majus* with *Ralstonia solanacearum* strains has never been demonstrated, being restricted to an artificial infection [15,16].

Even with the great interest in all these features observed throughout the *T. majus* plant as a whole, the plant-associated microbiome is yet very little known and/or explored. Several bacteria are known to promote growth of different plants (but still unknown in *T. majus*) and are generally designated as plant growth-promoting bacteria (PGPB). The beneficial effects of these bacteria on plant growth can be direct or indirect, such as increasing nutrient and water availability, enhancing abiotic and biotic stress tolerance and protecting against phytopathogens [17,18,19,20]. Therefore, the selection and further use of these PGPB as biofertilizers can contribute to a more sustainable agriculture, minimizing the use of synthetic fertilizers and agrochemicals [17].

Based on the unexplored potential of the *T. majus* bacteriome and the need to explore alternatives to promote plant growth and/or protect against phytopathogens, this study aims to characterize the total bacterial community associated with the rhizosphere, endosphere and bulk soil of *T. majus* through 16S rRNA amplicon metagenomic sequencing. Furthermore, we isolated endophytic bacteria (bacterial strains that live inside the plants tissue without causing damage [20]) to select and identify possible PGPB for the development of a biofertilizer in the near future.

## 2. Materials and Methods

### 2.1. Site Description and Experimental Design

The sampling was performed on 17 October 2019, at Sítio Cultivar, a 42 hectare organic farm with an altitude of 1067 m located in Nova Friburgo, Rio de Janeiro (22°17′53″ S/42°27′35″ W). The region has an average annual precipitation and temperature of 2174 mm and 18.14 °C, respectively. The plants were irrigated twice a day and fertilized every three months with a fermented organic fertilizer (Fert-Bokashi^®^, Korin, Brazil).

Five different *T. majus* plants were harvested, and the roots were shaken to remove the loosely attached soil. The adhering soil of each plant (500 mg) was taken with a sterile spatula and considered rhizospheric soil. The roots of the different plants were transported to the laboratory separately in sterile bags. The bulk soil was also sampled in five different spots (depth of 0–10 cm). The physicochemical characteristics of the soil where the *T. majus* plants were planted (a loamy soil with medium texture) are shown in Appendix A.

### 2.2. Isolation of Endophytic Bacteria

Approximately 10 g of roots from each of the five plants was individually weighed and washed with 15 mL of sterile saline (NaCl 0.85%) under agitation (100 rpm) at 28 °C for 1 h. Root samples were then surface disinfected with UV light exposure for 5 min before rinsing with 70% ethanol for 2 min and 2.5% sodium hypochlorite for 5 min and then washing three times with sterile distilled water. To check the efficiency of the disinfection procedure, 100 μL of the water used in the last wash was plated onto trypticase soy broth (TSB) agar-containing plates (TSA). Root samples that were not contaminated according to the culture-dependent sterility test were homogenized with 15 mL of sterile distilled water in a sterilized mortar and pestle and used for the isolation of endophytic bacterial strains and for DNA isolation.

Disinfected root samples were successively diluted (10^0^ to 10^−8^) in sterile saline, and 100 µL of each dilution was inoculated in triplicate in TSA and incubated at 32 °C for up to 5 days. Different colonies were selected based on their morphotypes. The different isolates were maintained at −80 °C in TSB with 20% glycerol.

### 2.3. DNA Extraction of Bulk Soil, Rhizosphere and Endosphere

DNA extraction from 500 mg of the root macerate (endosphere), rhizospheric soil (rhizosphere) and bulk soil—totalizing 15 samples—was performed using the DNeasy PowerSoil kit (Qiagen^®^, Hilden, Germany) following the manufacturer’s instructions. The resulting DNA samples were quantified (ng/μL) using Qubit™ dsDNA HS Assay Kit (Thermo Fisher Scientific^TM^, Waltham, MA, USA), and their integrity was checked through agarose gel electrophoresis (0.8%) with SYBR™ Safe DNA Gel Stain (3%), prepared in TBE 1X buffer [21] for 2 h at 80 V.

### 2.4. Amplicon Metagenomic Sequencing of 16S rRNA Genes from Bulk Soil, Rhizosphere and Endosphere

Total DNA extracted (approximately 20–40 ng/μL) from the 15 samples (endosphere, rhizosphere and bulk soil) was sent to Novogene (Sacramento, CA, USA) and sequenced on the Illumina NovaSeq 6000 platform, as recommended by the manufacturer. The primers 515F and 806R [22] with the addition of barcodes were used to amplify fragments of 300 bp from the V4 region of the *rrs* gene (encoding 16S rRNA). Paired-end sequencing libraries (2 × 250 bp) were constructed. Finally, the data were filtered by removing the barcodes and primers from the raw reads to obtain high-quality sequences.

### 2.5. Bioinformatic Analysis

The obtained sequences (15 samples) were analyzed with Mothur v.1.43.0 software [23]. Forward and reverse sequences were paired in contigs, and homopolymers (≥8), ambiguities and sequences with inconsistent sizes were removed, while identical sequences were grouped. Virtual PCR was performed with the primers 341F and 806R [24] to align the remaining sequences with the Silva v.138 database [25]. Then, the number of sequences per sample was rarified using the lowest number of sequences obtained in a sample, preclusters were built, and chimeric sequences were removed. The sequences were classified using the Ribosomal Database Project (RDP [26]), and possible contaminants were removed, such as mitochondrial DNA, chloroplasts, Archaea, Eukarya and unknown taxa (cut off = 80%). Similar sequences were classified into operational taxonomic units (OTUs) using the “cluster.split” command (cut off = 1%). Finally, the data related to alpha and beta-diversity indexes (OTUs, Chao1, Shannon and Simpson indexes), rarefaction curves and taxonomic relative abundance were used in further statistical analyses.

### 2.6. Statistical Analyses

PAST v4.02 software [27] was used in the statistical analyses. The diversity and taxonomic relative abundance data were submitted to Shapiro–Wilk normality test, and the resulting data were transformed using Box–Cox or Log transformation [28] when necessary (*p* < 0.05). The data showing a normal distribution (parametric) were submitted to one-way ANOVA followed by Tukey’s test to evaluate which sites (endosphere, rhizosphere and bulk soil) showed a significant difference among themselves (*p* < 0.05). The nonparametric data were submitted to the Kruskal–Wallis test followed by the Mann–Whitney U test.

The distribution matrix of OTUs was submitted to nonmetric dimensional scaling (NMDS) using the Bray–Curtis dissimilarity index to evaluate the distribution and correlation of OTUs between the sampled sites. PERMANOVA test was performed to analyze the significant differences between the sampled sites (*p* < 0.05). Finally, the OTU distribution matrix with its respective taxonomic identifications was submitted to a linear discriminant analysis Effect Size (LEfSe) to evaluate which taxonomic groups were significantly enriched (*p* < 0.05) in each sampled site, the consistency of these results among the five replicates of each site, and the possible relevance of this effect [29]. LEfSe was performed in the Huttenhower Lab online platform with the Galaxy Community Hub server.

### 2.7. Endosphere PGPB Selection

The different *T. majus* endophytes were grown in TSB (3 mL) for 24 h at 32 °C and used in the different plant growth-promoting tests listed below.

#### 2.7.1. Production of Indole-Related Compounds

Each bacterial culture (100 µL) was inoculated in 3 mL of King’s B medium [30] and incubated for 72 h in the dark at 27 °C under agitation (150 rpm). According to the method described by Tang and Bonner [31], the culture supernatants (1 mL) were mixed with 1 mL of Salkowski reagent (1.875 g FeCl_3_.6H_2_O, 100 mL H_2_O and 150 mL H_2_SO_4_ at 96% purity). The presence of indole-related compounds (IRCs) was considered when the color of the culture supernatant became reddish.

#### 2.7.2. Organic Phosphate Mineralization and Inorganic Phosphate Solubilization

Inorganic phosphate solubilization (PS) tests were carried out in NBRIP agar-containing plates [32] supplemented with calcium phosphate. Organic phosphate mineralization (PM) tests were performed as described in Rosado et al. [33] using calcium phytate as the phosphorus (P) source. The bacterial strains were inoculated in both media as 5 μL spots, and the tests were considered positive whenever a clear zone (halo) around the bacterial growth was observed after 5 days at 32 °C.

#### 2.7.3. Siderophore Production

The bacterial strains were inoculated as spots (5 µL) in CAS-agar selective medium [34] and incubated at 32 °C for 5 days. Siderophore (SID) production was detected as described by Schwyn and Neilands [34], where the presence of a yellow halo around the colonies was considered a positive result.

#### 2.7.4. Production of Antimicrobial Substances

The overlay method described by Rosado and Seldin [35] was used to detect antimicrobial activity. All isolates were inoculated onto TSA plates as 5 μL spots, and after incubation at 32 °C for 48 h, the cells were killed by exposure to chloroform vapor for 30 min. The plates were then flooded with suspensions containing *Micrococcus* sp. as the indicator strain [36]. Antimicrobial substance (AMS) production was indicated by clear zones of inhibition observed around the spots after 24 h at 32 °C.

#### 2.7.5. Amplification of the Nitrogenase Encoding Gene

The *nifH* gene sequences were PCR amplified from bacterial colonies grown in TSA as described by Woodman [37]. The primers UEDA19F and R6 and the PCR conditions were those described in Angel et al. [38]. Negative controls (without DNA) were run in all amplifications. PCR products were analyzed by 1.4% agarose gel electrophoresis followed by staining with SYBR™ Safe to confirm their expected sizes (394 bp).

### 2.8. DNA Extraction of the Selected PGPB

Bacterial genomic DNA was extracted using the ZR Fungal/Bacterial DNA MiniPrepTM kit (Zymo Research Corporation, Irvine, CA, USA) according to the manufacturer’s protocol. The DNA was quantified spectrophotometrically as described above.

### 2.9. Molecular Identification of Selected PGPB

PCRs were performed using the DNA extracted from the selected PGPB and the universal primers pA and pH, as described by Massol-Deya et al. [39]. The amplification conditions were 94 °C for 2 min, followed by 35 cycles of 94 °C for 70 s, 48 °C for 30 s, and 72 °C for 10 s, and a single step of 72 °C for 2 min. The PCR products were purified using the commercial kit Wizard^®^ SV Gel and PCR Clean-Up System (Promega Corporation, Madison, WI, USA), following the manufacturer’s protocol.

Sequencing reactions were prepared with bacterial DNA (20–40 ng), primers (0.5 µM—785F or 907R [40]) and BigDye^TM^ Terminator v3.1 Cycle Sequencing Kit (Applied Biosystems—Thermo Fisher Scientific^TM^, Waltham, MA, USA), following the manufacturer’s instructions. The automatic sequencer SeqStudio^TM^ (Applied Biosystems—Thermo Fisher Scientific^TM^) was used.

The forward and reverse sequences of each strain obtained were initially analyzed by the Electropherogram quality analysis tool (asparagin.cenargen.embrapa.br/phph/ (accessed on 21 July 2021), thus removing low-quality bases/sequences (phred < 20). Afterwards, the contigs were assembled using BioEdit software [41] and submitted to the rRNA/ITS database in the National Center for Biotechnology Information (NCBI) through the BLASTn tool (Nucleotide Basic Local Alignment Search Tool) to compare the sequences obtained to the sequences deposited in those databases, focusing on the highest identities (>97%) and coverages (>99%).

### 2.10. Phylogenetic Analyses

Sequences of closely related bacterial strains were recovered from the GenBank database and aligned to the sequences obtained in this study using MEGA X software [42] to infer a possible evolutionary correlation between those bacteria. The sequences were aligned through the ClustalW method to analyze the similarity between those sequences. The phylogenetic tree was constructed using the maximum likelihood method and the Jukes–Cantor model (bootstrap = 500). Moreover, another phylogenetic tree was generated exclusively with the sequences from the isolated strains (bootstrap = 100). This second tree was exported to iTOL v6 [43] to include the metadata of the plant growth-promoting tests performed here and their possible molecular identification.

### 2.11. Comparison between the Total Endophytic Bacterial Community and the Selected PGPB

The file containing the sequences of the selected PGPB was imported into Mothur software v.1.43.0 [23] and was incorporated into the file containing the sequences of the endosphere bacterial community. All sequences were aligned with Silva database v.138 [25] through virtual PCR (align.seqs) using the primers 341F and 806R [24]. The sequences were filtered and clustered as described above.

## 3. Results

### 3.1. Endosphere, Rhizosphere and Bulk Soil Bacteriome Analyses through 16S rRNA Amplicon Metagenomic Sequencing

A total of 1,429,053 sequences were obtained from the 15 samples (bulk soil, rhizosphere and endosphere). The number of sequences was normalized to 49,071 per sample, totaling 736,065 sequences and 64,900 OTUs. The rarefaction curves shown in Appendix A indicate that the number of sequences obtained from the three sampling sites (endosphere, rhizosphere and bulk soil in triplicate) was enough to cover most of the local bacterial communities (bacteriomes). They also revealed that bulk soil presented greater sample richness than the other two sites (Appendix A).

The alpha diversity analyses showed that the bulk soil bacteriome had significantly higher (*p* < 0.05) richness (Chao1 index) and diversity (Shannon index) than the endosphere bacteriome, as shown in Figure 1A. Moreover, the rhizosphere bacteriome presented a significantly lower (*p* < 0.05) richness than the bulk soil bacteriome. The dominance of bacterial taxa (Simpson index) was significantly higher (*p* < 0.05) in the endosphere bacteriome than in the bulk soil bacteriome. Richness and diversity were higher and dominance was lower in the rhizosphere bacteriome than in the endosphere bacteriome; however, these results were not statistically significant.

The beta diversity analysis among the three bacteriomes (endosphere, rhizosphere and bulk soil) was represented in a nonmetric multidimensional scaling (NMDS) analysis (Figure 1B). The bacteriome from the bulk soil, rhizosphere and endosphere differed significantly (*p* < 0.05), demonstrating the influence of the sites in the grouping of samples. PERMANOVA statistically confirmed this observation.

To gain better knowledge of the bacteriome composition, the relative abundance of bacterial taxa in the three sampling sites was determined. OTUs related to 15 phyla were found in all sampled sites (in different proportions): Proteobacteria, Actinobacteria, Firmicutes, Acidobacteria, Bacteroidetes, Plantomycetes, Verrucomicrobia, Gemmatimonadetes, Nitrospirae, Chloroflexi, Armatimonadetes, Latescibacteria, candidate division WPS-1, Spirochaetes and Deinococcus-Thermus. The eight phyla with at least 1% relative abundance in the bulk soil, rhizosphere and/or endosphere bacteriomes are shown in Appendix A.

The phylum Proteobacteria predominated in the three sites (rhizosphere—60.9%, endosphere—54.6% and bulk soil—41.8%). The bulk soil bacteriome showed a significantly higher (*p* < 0.05) relative abundance of the phyla Actinobacteria, Acidobacteria and Verrucomicrobia (19.2%, 9.4% and 2.12%, respectively) when compared to the rhizosphere (10.5%, 4.8% and 1.1%, respectively) and endosphere (8.62%, 3.87% and 1%, respectively) (Appendix A).

The 20 most abundant genera, *Ralstonia*, *Pseudomonas*, *Paenibacillus*, *Gemmatimonas*, *Gaiella*, *Rhizobium*, *Mycobacterium*, *Yersinia*, *Sphingomonas*, *Novosphingobium*, *Sediminibacterium*, *Nocardioides*, *Acidovorax*, *Nitrospira*, *Streptomyces*, *Dyella*, *Stenotrophomonas*, *Solirubrobacter*, *Rhodobacter* and *Serratia*, showed a relative abundance of at least 0.75% in the different bacteriomes (Figure 2). Altogether, they represented 32.19% of the total bacterial community obtained from the three sites (Figure 2).

The bacteriome associated with bulk soil showed a higher relative abundance (*p* < 0.05) of some genera from the phylum Actinobacteria, such as *Gaiella*, *Nocardioides*, *Streptomyces* and *Solirubrobacter*, when compared with those of the endosphere and rhizosphere. Moreover, the rhizosphere bacteriome had a significantly higher (*p* < 0.05) relative abundance of the genera *Rhizobium* and *Sphingomonas* than the endosphere bacteriome and of the genus *Pseudomonas* than the endosphere and bulk soil. The endosphere bacteriome had a significantly higher (*p* < 0.05) relative abundance of the genera *Paenibacillus* and *Ralstonia* when compared to the rhizosphere and bulk soil bacteriomes, respectively (Figure 2). The relative abundance of other genera also varied (significantly higher or lower) when the different sites were compared (Figure 2).

Additionally, LEfSe was performed to compare the richness of OTUs in the bacteriome of the different sites, considering the relative abundance and the consistency of these data between the replicates, excluding possible outliers (Figure 3). The bacteriome associated with bulk soil was significantly enriched (*p* < 0.05) in bacteria from the order Actinomycetales (phylum Actinobacteria). In addition, the endosphere bacteriome was significantly enriched with strains from Proteobacteria, especially from the classes Alphaproteobacteria (e.g., Rhodospirillaceae) and Betaproteobacteria (e.g., *Schlegelella* sp.) when compared to the other sites. Finally, the bacteriome associated with the rhizosphere presented itself as an interface area between the bulk soil and endosphere, as it was significantly enriched with bacteria from the phyla Proteobacteria (e.g., *Paracoccus* sp. and *Mesorhizobium* sp.), Acidobacteria (e.g., Acidobacteria_Gp4) and Actinobacteria (e.g., Promicromonosporaceae).

### 3.2. Selection and Identification of Plant Growth-Promoting Bacteria (PGPB)

A total of 236 bacterial strains were isolated from the *T. majus* endosphere of the five different plants sampled. These strains were submitted to six different plant growth-promoting tests: phosphate mineralization (PM), phosphate solubilization (PS), siderophore production (SID), production of antimicrobial substances (AMS), production of indole-related compounds (IRCs) and presence of the *nifH* gene (*nifH*). Appendix A shows the results of the 236 isolates for each test performed. Approximately 95% of the strains were positive for at least one of the plant growth-promoting tests performed, while only 5% of them were positive for all tests. In addition, 64.8% strains were positive for PM, 70.3% for PS, 66.9% for SID, 64.8% for AMS, 44.9% for IRCs and 13.1% for *nifH* (Appendix A). Strains that were positive for at least four out of six plant growth-promoting tests were further molecularly identified.

Of the 101 strains chosen for 16S rRNA gene sequencing, 25 resulted in DNA fragments below 1000 bp and were discarded (Appendix A). The 76 sequences ranging from 1003 to 1500 bp were submitted to the BLASTn database and identified as belonging to the phyla Actinobacteria, Firmicutes and Proteobacteria. Eleven different genera were identified: *Mycolicibacterium*, *Bacillus*, *Paenibacillus*, *Staphylococcus*, *Pseudomonas*, *Pantoea*, *Enterobacter*, *Citrobacter*, *Klebsiella*, *Raoultella* and *Serratia* (Appendix A).

These 76 DNA sequences were also exported to MEGA X to construct a phylogenetic tree (Figure 4) considering the similarities between these sequences. The metadata results related to plant growth-promoting tests and the closest taxonomic identifications were included in the tree.

The cluster *Raoultella*/*Klebsiella* (Figure 4—green branches) showed a similar phenotypic profile, including the PGPB that were positive for six plant growth-promoting tests performed. In contrast, a diverse phenotypic profile was observed when the five strains with high identities to the genus *Bacillus* were considered. As they could be observed spread in the phylogenetic tree, we suggest that those strains belong to more than one *Bacillus* species.

Two other phylogenetic clusters were noticeable, and they included the strains with identity to the *Serratia* genus. The first cluster was formed by 21 strains (Figure 4—purple branches), and the second cluster was formed by 17 strains (Figure 4—yellow branches). These two *Serratia* clusters showed a variable phenotypic profile and a phylogenetic distance between them, suggesting that they belong to different species of *Serratia*.

The 16S rRNA sequences from the isolated strains and those of the bacteria deposited in the databases (BLASTn) were also used to construct other phylogenetic trees. The first three hits with the highest identities (>97%) were selected for each bacterial strain isolated here, and their accession numbers are shown in Appendix A. These phylogenetic trees corroborate the findings stated above. Five isolated strains showed high similarity to the genus *Bacillus* (Appendix A), 13 isolates to *Raoultella*/*Klebsiella* (Appendix A), and 21 isolates clustered with *Serratia entomophila*, *S. ficaria*, *S. plymuthica*, *S. liquefaciens* and *S. grimesii*, whereas 17 strains clustered with *S. nematodiphila*, *S. surfactantfaciens*, *S. marcescens* and *S. ureilytica* (Appendix A).

### 3.3. Comparison between the Endosphere Bacteriome and the Bacterial Isolates

The 16S rRNA sequences obtained with the culture-dependent and culture-independent methods were merged and analyzed. Five OTUs were shared between the sequences from the endosphere bacteriome and the isolated endophytic strains (Figure 5). Approximately 10,000 and 2000 sequences were clustered in OTU4 and OTU12, respectively, which comprise the *Bacillus* genus and include the sequences from the strains E45 and E49 (OTU4) and from the strains E92 and E95 (OTU12). Moreover, 5357 sequences were clustered in OTU6 (*Pseudomonas* genus) and included the strain E69 sequence. Finally, 327 sequences were clustered in OTU123 (*Staphylococcus* genus) including strain E12, and 1772 sequences clustered in OTU15 (*Serratia* genus) including the sequences from 16 isolated strains (Figure 5).

## 4. Discussion

An initial characterization of plant-associated bacteriomes following culture-dependent (isolation and characterization of bacteria) and culture-independent approaches (total bacterial communities) provides new insights into the plant microbiome profile and represents a first step into a potentially promising strategy for the identification of prospective plant growth-promoting bacterial agents well adapted to the ecological niche in which these organisms would be potentially applied. These approaches have already been used in different studies [44,45,46]. Additionally, in this study, the total bacterial communities associated with the rhizosphere, endosphere and bulk soil of *T. majus* were molecularly determined (their structure and composition), and endophytic bacteria were isolated and further characterized to determine their plant growth-promoting potential.

With the data obtained from the 16S rRNA sequencing of the *T. majus* bacteriome from the bulk soil, rhizosphere and endosphere, the alpha diversity analyses showed a gradual reduction in bacteriome richness and diversity from the bulk soil to the rhizosphere and from the rhizosphere to the endosphere. Similar results were obtained with different plants, such as *Glaux maritima*, *Lolium perenne* and *Trifolium repens* [47,48,49,50]. It seems that the more intimate the interaction between plants and bacteria, the more specialized these bacteria need to be so they can colonize the plant.

The most abundant OTUs associated with *T. majus* (endosphere, rhizosphere and bulk soil) were from Proteobacteria. The phylum Proteobacteria includes free-living bacteria, phytopathogens and potential PGPB [51]. Within this phylum, the genera *Pseudomonas* and *Ralstonia* predominated considering the three sites analyzed. While *Pseudomonas* strains can promote plant growth through diverse mechanisms, such as acting as a biocontrol agent [52], strains belonging to the *Ralstonia* genus comprise phytopathogenic species, currently clustered in the *Ralstonia solanacearum* species complex (RSSC), able to affect more than 200 plant species [53].

Healthy plants usually present a relative abundance of *Ralstonia* under 1% [54]. Previous studies in tomato plants infected with *R. solanacearum* showed a relative abundance of the *Ralstonia* genus in the rhizosphere next to 35%, while healthy plants showed a relative abundance of 0% [55]. To our knowledge, a natural infection of this phytopathogen in *T. majus* plants has not been described thus far. Therefore, more studies are necessary to determine the ecological interaction between *Ralstonia* and *T. majus* plants.

Despite the great amount of information obtained here from the culture-independent approach, we are aware that there are limitations inherent to molecular techniques, such as primer specificity [37] and limited information in the databases [56]. Therefore, we also used a culture-dependent method to recover microorganisms that may not be detected by 16S rRNA amplicon sequencing and that may be used in in vivo experiments.

We were able to isolate 76 strains from the endosphere of *T. majus* that were molecularly identified in 11 different genera. The great majority of the isolates were identified as belonging to the genera *Serratia* and *Raoultella*/*Klebsiella* (phylum Proteobacteria). *Raoutella* and *Klebsiella* isolates were phylogenetically very similar, according to previous observations [57,58].

Strains belonging to *Klebsiella* and *Serratia* have already been proposed to be used as biofertilizers, especially due to the capacity of certain strains to fix atmospheric nitrogen, solubilize phosphate, produce siderophores, produce indole-related compounds and/or produce antimicrobial substances [59,60]. These plant growth-promoting characteristics were also observed in many of our isolates.

Many strains belonging to the genera *Bacillus* and *Paenibacillus* (phylum Firmicutes) were also isolated in this study. *Bacillus* is one of the most extensively studied rhizobacteria promoting the growth of many crops [61]. Its application as biofertilizer is efficient due to the presence of diverse mechanisms of nutrient and hormone availability for plants, activity against phytopathogens and tolerance to drought and salinity stresses [62]. *Paenibacillus* strains are also potential candidates for application as biofertilizers, as they are capable of fixing atmospheric nitrogen, solubilizing phosphate, producing siderophores, producing exopolysaccharides, liberating phytohormones and other features [63]. One benefit of inoculating plants with endospore-forming bacteria such as *Bacillus* or *Paenibacillus* is their capacity to survive for long periods in the soil under adverse environmental conditions [64].

Finally, 16S rRNA sequences of the bacterial endophytes and the different OTUs of the endosphere bacteriome were compared. Twenty-two strains were consistently found in both approaches: *Bacillus* (E45, E49, E92, and E95), *Pseudomonas* (E69), *Serratia* (E17, E33, E34, E50, E52, E54, E55, E57, E64, E67, E75, E77, E81, E102, E114, and E126) and *Staphylococcus* (E12).

As previously discussed, the use of these isolated strains as biofertilizers may facilitate the establishment of these bacteria in *T. majus* plants. These potential biofertilizers may play an important role in maintaining the productivity and sustainability of soil systems. However, we are aware that these genera harbor species that are not purely beneficial, with some of them causing food spoilage or opportunistic infections in humans, representing a potential threat to human, animal or plant health [65]. Moreover, PGPB selected in the laboratory may fail to confer the expected beneficial effects when evaluated in plant experiments, caused by insufficient rhizosphere/endosphere colonization. Therefore, the effectiveness and security of the potential biofertilizers presented here have to be demonstrated through several experiments under greenhouse and field conditions, proving their ability to promote plant growth in *T. majus*.

## 5. Concluding Remarks

Culture-independent and culture-dependent methods were employed for the first time to analyze the bacteriome associated with *Tropaeolum majus* plants. Each of the three sites analyzed (bulk soil, rhizosphere and endosphere) associated with *T. majus* showed a specific bacterial community composition. A gradual reduction in bacteriome richness and diversity from the bulk soil to the rhizosphere and from the rhizosphere to the endosphere was observed. The taxa (OTUs) enriched in the *T. majus* rhizosphere and endosphere were mostly associated with biocontrol and nutrient availability to the plant, such as the genera *Pseudomonas* and *Paenibacillus*. Twenty-two endophytes isolated from *T. majus* associated with the genera *Bacillus*, *Pseudomonas*, *Staphylococcus* and *Serratia* were detected in both culture-dependent and molecular methods. Further studies are still necessary to better understand each bacterial strain isolated here and their mechanisms of action as PGPB to enhance *T. majus* health and growth in field conditions.

## Figures and Tables

**Figure 1 microorganisms-10-00638-f001:**
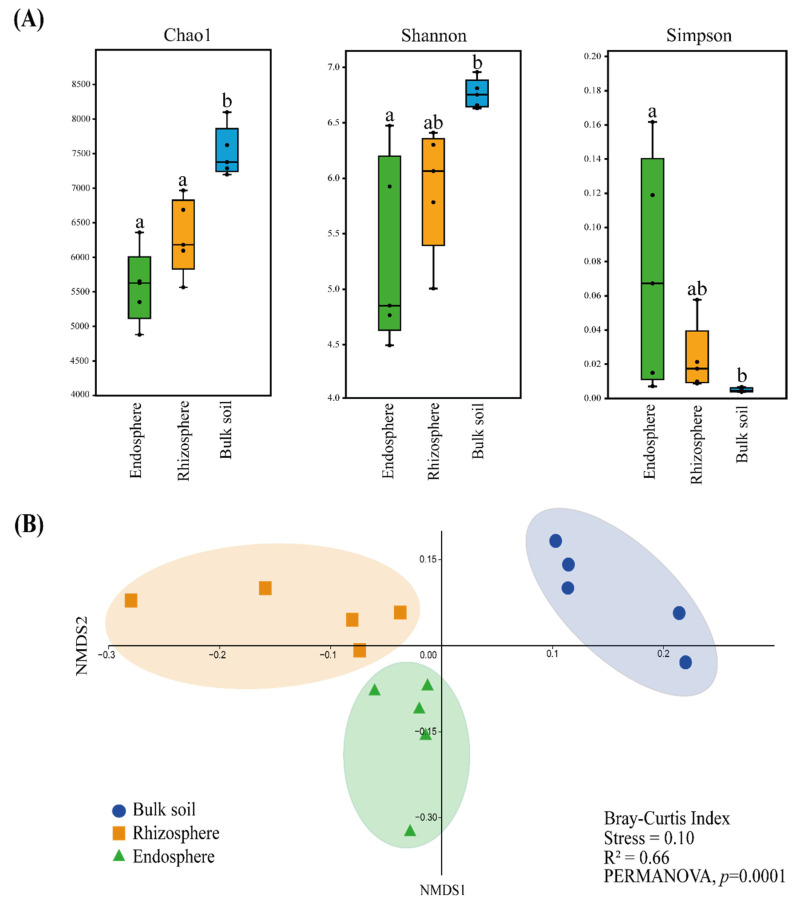
Diversity analyses of bulk soil (blue), rhizosphere (orange) and endosphere (green) bacteriomes associated with *Tropaeolum majus*. (**A**) Alpha diversity analyses. Different letters indicate significant differences (Tukey’s test, *p* < 0.05). (**B**) Beta diversity analysis represented in nonmetric multidimensional scaling (NMDS) generated in a 3D plot based on the Bray–Curtis dissimilarity index.

**Figure 2 microorganisms-10-00638-f002:**
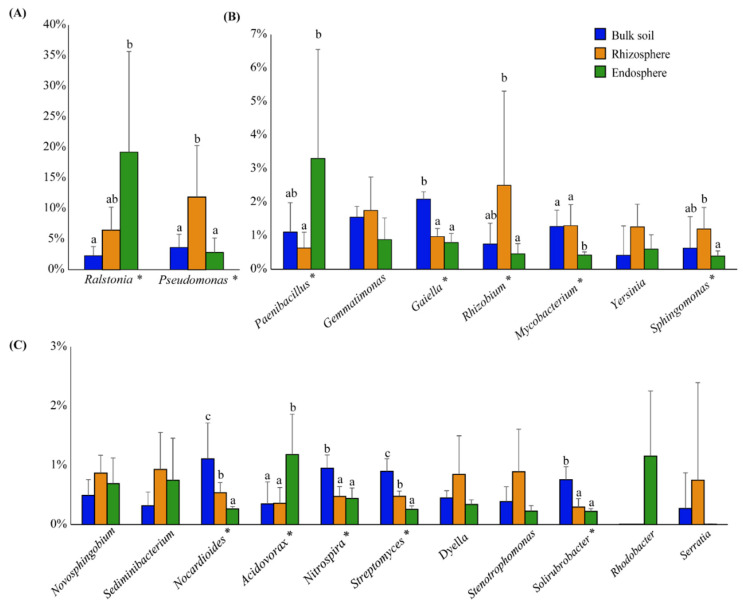
The relative abundance of bacterial genera associated with the bulk soil, rhizosphere and endosphere of *Tropaeolum majus* determined through 16S rRNA amplicon metagenomic sequencing. Asterisks next to the genus name indicate that there is a significant difference. Different letters above error bars indicate which sampled site showed significant differences (Tukey’s test or Mann–Whitney U test, *p* < 0.05). (**A**) Up to 40% relative abundance, (**B**) up to 7% relative abundance and (**C**) up to 3% relative abundance.

**Figure 3 microorganisms-10-00638-f003:**
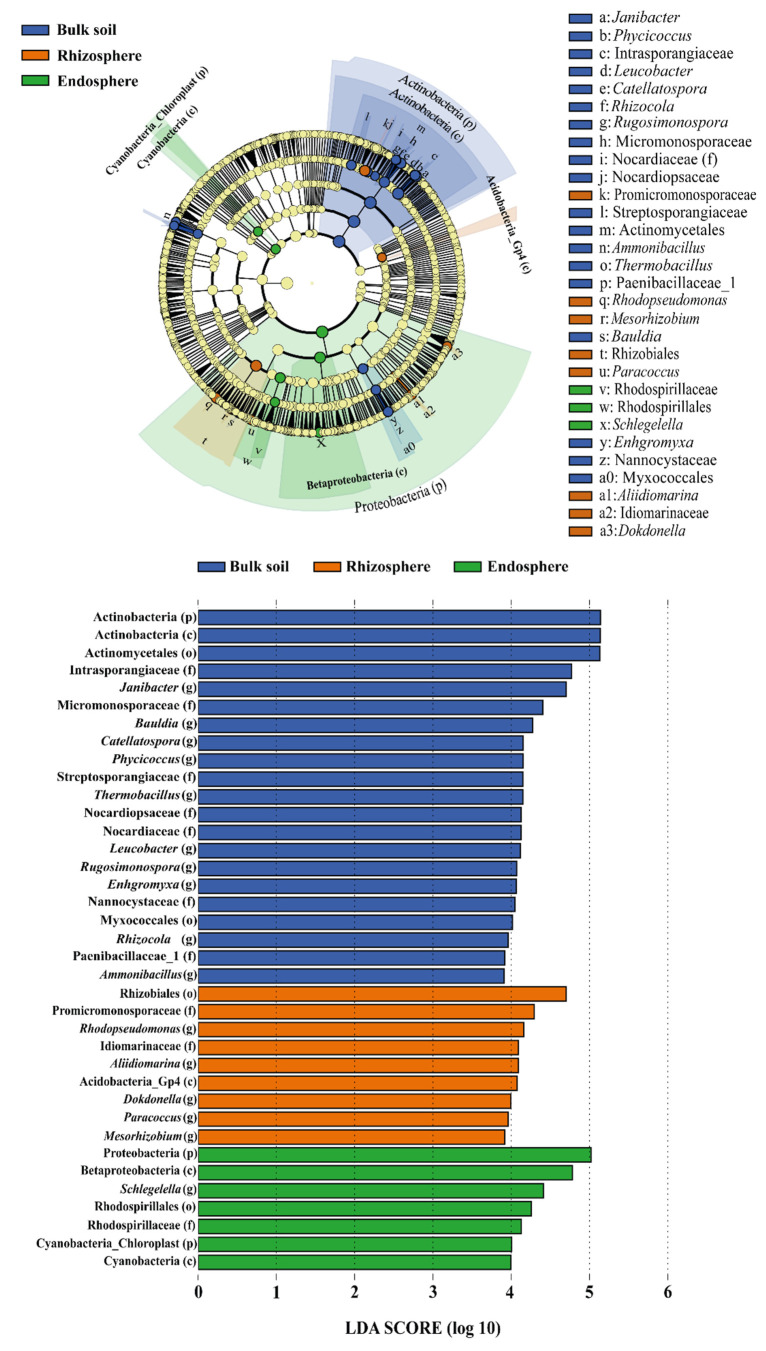
Linear discriminant analysis Effect Size (LEfSe) used to evaluate the significantly enriched OTUs in the bacteriomes of bulk soil (blue), rhizosphere (orange) and endosphere (green) associated with *Tropaeolum majus.* The analysis was performed in the Huttenhower Lab online platform with the Galaxy Community Hub server.

**Figure 4 microorganisms-10-00638-f004:**
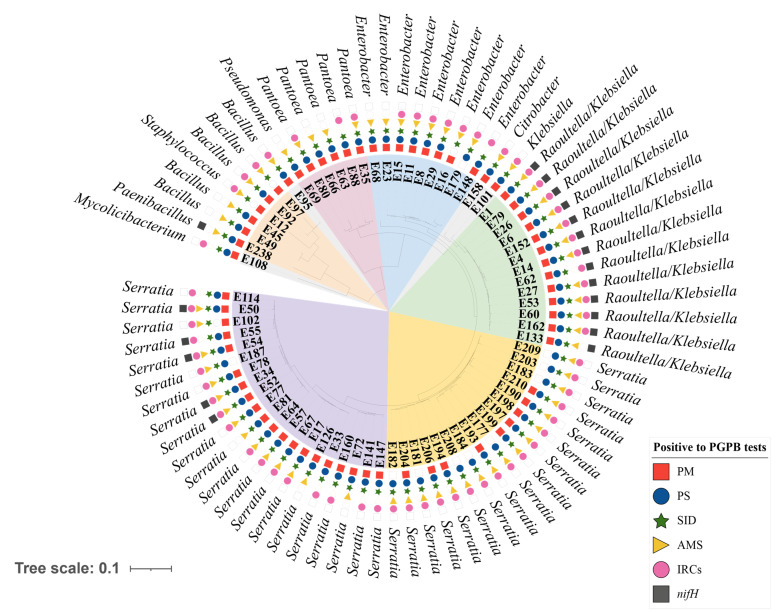
Phylogenetic tree of 16S rRNA sequences from bacterial strains isolated from the *Tropaeolum majus* endosphere, including the metadata related to the plant growth-promoting tests and their closest molecular identification. The phylogenetic tree was constructed with MEGA X software using the maximum likelihood method, and the metadata were added with iTOL v6 software. PM—phosphate mineralization, PS—phosphate solubilization, SID—siderophore production, AMS—production of antimicrobial substances, IRCs—production of indole-related compounds, and *nifH*—presence of the *nifH* gene.

**Figure 5 microorganisms-10-00638-f005:**
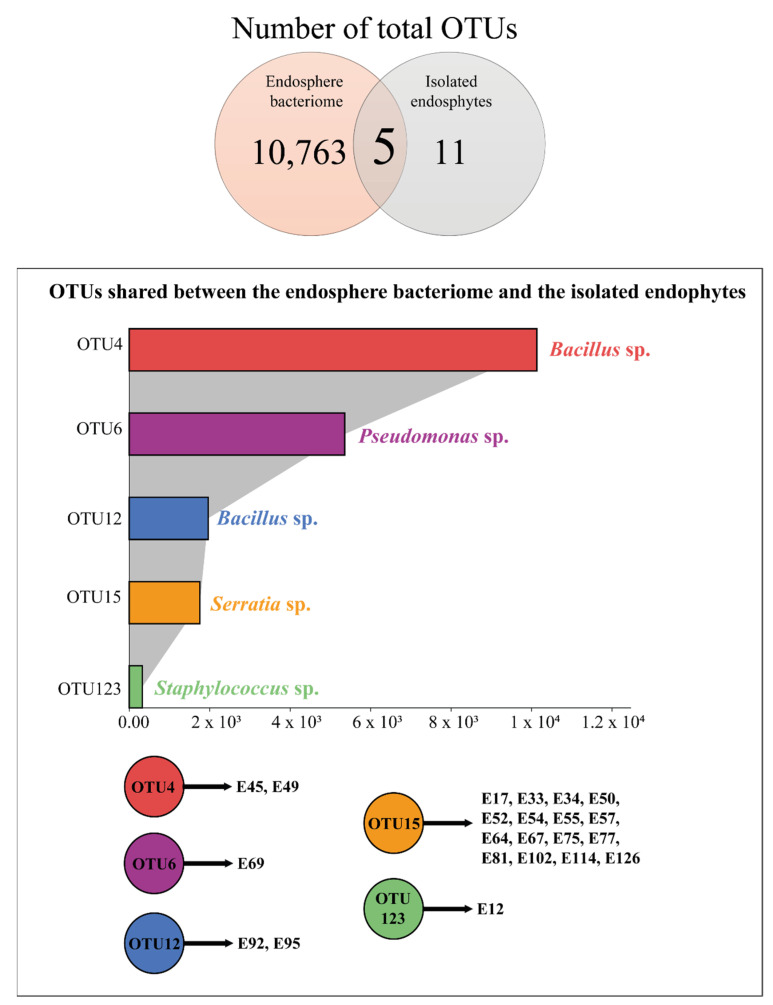
Comparison between the 16S rRNA sequences obtained from endosphere amplicon metagenomic sequencing and the sequences from the isolated endophytic PGPB. The five OTUs shared, as well as their identification at the genus level and the total number of sequences of each OTU, are detailed. The endophytic bacterial sequences clustered within the five OTUs are also listed.

## Data Availability

The data presented in this study are available upon request to the corresponding author (L.S.).

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
