# Peer review of "Unraveling the Tropaeolum majus L. (Nasturtium) Root-Associated Bacterial Community in Search of Potential Biofertilizers"

_microorganisms, 2022, doi:10.3390/microorganisms10030638_

Round 1

Reviewer 1 Report

The manuscript is well-written, I recommend to accept it.

Author Response

The authors thank the reviewer for his/her complimentary words about the revised manuscript.

Reviewer 2 Report

Dear authors.

The potential of plants and microorganisms for ecosystem restoration is widely studied. Despite active research in this area, the mechanism of action in each case is individual and requires detailed study. The topic touched upon in the article is relevant. The scientific content of the manuscript justifies its publication, but some additions and modifications will significantly improve the quality of the article.

Major comments:

1) L.73. , "endophytic bacteria" should be determined before use.

2) L.228, 363, p. 3.1 should be corrected.

3) L. 386, [44-46, and others], "others" require an explanation.

4) The mechanism of selection of trains for the study of their suitability in biofertilizers is not clear. The authors need to add their vision of this issue.

5) In the References, 31% of publications refer to 2016-2020 (the last 5 years); the remaining 69% of used sources are older than 5 years. It is recommended to increase the share of references to sources published over the last 5 years when analyzing the current state of research in the area under consideration, since this area of knowledge is rapidly developing.

Author Response

Reviewer 2: Major comments:

1) L.73. , "endophytic bacteria" should be determined before use.

Response: the term "endophytic bacteria" was explained. The sentence is now: Furthermore, we isolated endophytic bacteria (bacterial strains that live inside the plants tissue without causing damage [20]) to select and identify possible PGPB for the development of a biofertilizer in the near future.

2) L.228, 363, p. 3.1 should be corrected.

Response: In our manuscript version, the item was correct. However, the built PDF version was altered, and we will request to correct it.

3) L. 386, [44-46, and others], "others" require an explanation.

Response: As we did not mention the other references, it was deleted - [44-46]

4) The mechanism of selection of trains for the study of their suitability in biofertilizers is not clear. The authors need to add their vision of this issue.

Response: The last paragraph of the discussion was rewritten to make it clearer.

5) In the References, 31% of publications refer to 2016-2020 (the last 5 years); the remaining 69% of used sources are older than 5 years. It is recommended to increase the share of references to sources published over the last 5 years when analyzing the current state of research in the area under consideration, since this area of knowledge is rapidly developing.

Response: Six references were substituted by more recent ones. Now, 43% of the citations refer to the last five years, and 63% to the last decade.